# Can AI Help Pediatricians? Diagnosing Kawasaki Disease Using DRSA

**DOI:** 10.3390/children8100929

**Published:** 2021-10-17

**Authors:** Bartosz Siewert, Jerzy Błaszczyński, Ewelina Gowin, Roman Słowiński, Jacek Wysocki

**Affiliations:** 1Department of Preventive Health, Poznan University of Medical Science, 60-781 Poznan, Poland; ewego@ump.edu.pl (E.G.); jwysocki@ump.edu.pl (J.W.); 2Infectious Diseases Ward, Children’s Hospital in Poznan, 63-734 Poznan, Poland; 3Institute of Computing Science, Poznań University of Technology, 60-965 Poznan, Poland; jurek.blaszczynski@cs.put.poznan.pl (J.B.); roman.slowinski@put.poznan.pl (R.S.); 4Systems Research Institute, Polish Academy of Sciences, 01-447, Warsaw, Poland

**Keywords:** Kawasaki disease, infectious mononucleosis, *Streptococcus pyogenes*, children, DRSA, differential diagnosis

## Abstract

The DRSA method (dominance-based rough set approach) was used to create decision-making rules based on the results of physical examination and additional laboratory tests in the differential diagnosis of Kawasaki disease (KD), infectious mononucleosis and *S. pyogenes* pharyngitis in children. The study was conducted retrospectively. The search was based on the ICD-10 (International Classification of Diseases) codes of final diagnosis. Demographic and laboratory data from one Polish hospital (Poznan) were collected. Traditional statistical methods and the DRSA method were applied in data analysis. The algorithm formed 45 decision rules recognizing KD. The rules with the highest sensitivity (number of false negatives equals zero) were based on the presence of conjunctivitis and CRP (C-reactive Protein) ≥ 40.1 mg/L, thrombocytosis and ESR (Erythrocyte Sedimentation Rate) ≥ 77 mm/h; fair general condition and fever ≥ 5 days and rash; fair general condition and fever ≥ 5 days and conjunctivitis; fever ≥ 5 days and rash and CRP ≥ 7.05 mg/L. The DRSA analysis may be helpful in diagnosing KD at an early stage of the disease. It can be used even with a small amount of clinical or laboratory data.

## 1. Introduction

Despite the development of medicine, in some cases, there are still no tests giving a clear diagnosis. One of such examples is the differential diagnosis of Kawasaki disease (KD), infectious mononucleosis and Streptococcus pyogenes infections (angina, scarlet fever). Dealing with each of those diseases is completely different. Proper diagnosis and accurate treatment implementation, especially in KD and streptococcal pharyngitis, can reduce the risk of permanent complications [1].

Symptoms presented by the patient such as pharyngitis with cervical lymphadenopathy and ambiguous skin lesions are almost the same in each of those cases and do not differentiate the mentioned diseases. Additional tests might be helpful, but none of them are 100% sensitive. In Table 1, there are presented symptoms and laboratory findings at the initial stage of three analyzed diseases.

That is why we decided to use artificial intelligence to try to deal with this problem. In this study, for the first time, the original methodology of the dominance-based rough set approach (DRSA) was used to create decision-making rules based on the results of physical examination as well as additional tests that can potentially be helpful in the differential diagnosis of KD, infectious mononucleosis and *S. pyogenes* pharyngitis in children. DRSA enables testing, even in the case of incomplete data, enables the comparison of parametric and non-parametric data. The disadvantage of the classical scaling system is that the conversion of clinical data into numeric values risks losing the primary character of the data. Different parameters are added together as if they were equal, such that the sum of completely different parameters can yield the same results. The obtained results are non-informative in that they do not show how the diagnosis was made. It discourages the application of this system in therapeutic decisions because it does not give decision makers the chance to evaluate the independent results.

## 2. Materials and Methods

The study was conducted retrospectively. The search was based on the ICD-10 (International Classification of Diseases) codes of final diagnosis.

Data was collected from 1 January 2015 to 31 December 2019 in the Children’s Hospital in Poznan, which includes four pediatric departments. Children of both sexes up to five years of age in the case of infectious mononucleosis and *S. pyogenes* infection were included in the study. When considering KD, all children with this particular diagnosis were included in order to extend the study group as much as possible. The definite diagnosis of KD was based on AHA (American Heart Association) criteria. We used this age restriction because we wanted to eliminate adolescent patients with infectious mononucleosis. Epidemiological data show that this disease is most common in older children and young adults, in whom KD is very unlikely [2]. The second aspect is the prevalence of KD, primarily in children under five years of age. In the absence of an age restriction (excluding KD), comparing such inhomogeneous in age groups would not fit into the overall study aim.

Streptococcal pharyngitis was diagnosed based on positive rapid antigen tests for *S. pyogenes*. According to Polish recommendations, when there is a positive high-sensitivity test, it is not necessary to perform a throat culture [3]. The test was performed at the admission to the hospital, so at the beginning of diagnostic process, as well as before treatment started. The diagnosis of infectious mononucleosis was made based on positive serological results—positive IgM and IgG anti-VCA (anti-viral capsid antigen) antibodies. We collected following data: basic demographic information, the duration of the fever and physical examination results. General status was defined as good, fair, serious or critical. When it comes to examination, we chose those clinical signs which are known as diagnostic criteria for KD: general condition, presence of conjunctivitis, oral mucosa findings, distal extremities changes, skin changes and presence of cervical lymph nodes enlargement.

The laboratory test included those which are easily available: general blood count, alanine aminotransferase (ALT) activity, aspartate aminotransferase (AST) activity, urinalysis, urine culture, erythrocyte sedimentation rate (ESR), C-reactive protein (CRP) and procalcitonin levels. The values of the blood count parameters and the activity of ALT and AST were analyzed depending on the normal values specified for a given age group [4,5,6]. Blood for laboratory tests was collected during children’s admission.

### 2.1. DRSA Analysis

There were two subclasses among the data. The first included patients who were diagnosed with Kawasaki Disease. The second one contained the remaining patients, i.e., those who were finally diagnosed with infectious mononucleosis or Streptococcal infection. A relatively high number of missing values was observed for ESR and procalcitonin attributes. Overall, the resulting data set was fully consistent.

DRSA analysis was performed using jRS library and jMAF software package (http://www.cs.put.poznan.pl/jblaszczynski/Site/jRS.html (accessed on 16 October 2021).

Some attributes were transformed in a way that allowed the discover global and local monotonic relationships between condition and decision attributes [7]. The applied transformation was non-invasive, that is, it did not bias the matter of discovered relationships. Sets of decision rules, which were essential for this analysis, were induced using VC-DOMLEM algorithm [8]. These sets of rules were used to construct component classifiers in variable consistency bagging [9,10]. Variable consistency bagging (VC-bagging) and its variants developed to handle class imbalance were applied to increase the accuracy of results produced by VC-DOMLEM [11]. Estimation of attribute relevance in rules was performed according to Blaszczynski et al. [12]. All results were obtained in a 10-fold cross validation experiment that was repeated 20 times.

Most relevant rules with respect to correct classification of cases by decision rules in repeated cross validation are presented in Table 2. Correct classification to any of the two considered classes were investigated in this setting.

### 2.2. Statistical Analysis

Statistical analysis was performed using PQStat 1.8.0. (PQStatSoftware). For one group of data, Wilcoxon signed-rank test and χ^2^ test were used. Comparison of two groups were performed using the χ^2^ test, Student’s t-test or a correction for this test (Cochran–Cox). Shapiro–Wilk test was used to check normal distribution of data and Fisher–Snedecor test determined the equality of variance of variables in the analyzed population. While comparing three groups of data the Shapiro–Wilk test was used as well. Levene’s test was used to check whether the variances of the variables in the population are equal for the interval scale. When comparing groups in the interval scale, the ANOVA test was used for unrelated variables (mean age of patients, mean length of hospitalization). The χ^2^ test was used to compare the groups on the nominal scale. Post-hoc analysis, if needed, was performed with the Fisher test. For the SR value (Sedimentation Rate, Biernacki’s test), it was impossible to perform statistical analysis because of too few patients with this result. Statistical significance was calculated for PCT (procalcitonin) and CRP (C-reactive protein) values. The significance level was set at *p* < 0.05.

## 3. Results

There were 150 patients: 48 with KD, 49 patients with infectious mononucleosis and 53 with *S. pyogenes* pharyngitis. The analyzed population is characterized in Table 3.

Patients diagnosed with KD were admitted mainly during the winter and summer months (33% of all KD patients), those with infectious mononucleosis mostly during autumn months (31% of all these patients, *p* < 0.01) while children with *S. pyogenes* infection mostly in the spring (30% of the study group) and winter months (28% of the study group).

Patients with KD usually presented in a fair general condition (92%), while most patients with infectious mononucleosis (80%) and *S. pyogenes* pharyngitis (70%) were admitted in good general condition (*p* < 0.01).

When analyzing the diagnostic criteria for KD, 79% of the study group presented bilateral, nonexudative conjunctivitis; 90% presented changes on mucous membranes; 48% presented changes on the extremities (mainly edema); 83% presented with a rash; and 77% presented with cervical lymphadenopathy. Those KD symptoms were presented by patients with infectious mononucleosis, *S. pyogenes* pharyngitis as well as in patients with KD.

When it comes to laboratory tests, the most common abnormality among patients with KD were anemia (63%), thrombocythemia (65%), aseptic leukocyturia (40%) and elevated ALT and AST activity (48 and 42%, respectively).

In patients with infectious mononucleosis, anemia (14%) and elevated ALT and AST activity (27% and 33%, respectively) were observed less frequently, while thrombocytopenia and sterile leukocyturia were not observed at all.

Similarly, in patients with *S. pyogenes* pharyngitis, there were almost no cases of anemia or thrombocythemia (2% of patients in both groups). Aseptic leukocyturia occurred in 9% of patients, and transaminases activities were elevated in 8% of cases. The mean CRP and procalcitonin levels were increased in patients in all studied groups. The highest CRP results were observed in patients with KD (112.3 mg/L, *p* < 0.01), while procalcitonin was the highest in patients with *S. pyogenes* pharyngitis (2.37 µg/L, *p* = 0.44).

### DRSA Results

The most important predictors for the decision rules are presented in Figure 1.

Lymphadenopathy (1)—one side enlargement of cervical lymph nodes; Lymphadenopathy (2)—symmetrical (both sides) enlargement of cervical lymph nodes

The algorithm generated 45 decision rules recognizing KD. The rules with the highest sensitivity (number of false negatives equals zero) are as follows:If a child suspected of KD has conjunctivitis and CRP ≥ 40.1 mg/L, it is KD.If a child suspected of KD has thrombocythemia and ESR ≥ 77 mm/h, it is KD.If a child suspected of KD has a fair general condition, rash and fever lasting ≥ 5 days, it is KD.If a child suspected of KD has a fair general condition, conjunctivitis and fever lasting ≥ 5 days, it is KD.If a child with suspected KD has fever lasting ≥5 days, rash and CRP ≥ 7.05 mg/L, it is KD.

## 4. Discussion

Five of the rules generated using the DRSA method with the most clinical significance were based on following parameters: fair general condition, fever lasting ≥ 5 days, rash, conjunctivitis, CRP and ESR thrombocytosis. One of the big advantages of DRSA is using a combination of parametrical and non-parametrical variables.

Rules 3 and 4 were based on the results of physical examination only. What is more, both of them use patients’ general condition. The general condition assessed during admission to the hospital is a useful and easy-to-evaluate indicator, but it is still a subjective factor, depending on the experience of the doctor who makes this assessment [13,14]. The American Hospital Association has advised physicians to use general terminology when describing their patient’s state of health. The patient is Good when the patient’s vital signs are stable and they are within the normal limits. The patient is conscious, and the patient feels comfortable. The patient is Fair when the patient’s vital signs may be stable and within the prescribed normal limits. While the patient may be conscious, there may, however, be minor complications, making the patient somewhat uncomfortable. The patient is Serious when the patient’s vital signs might be fluctuating, and they do not adhere to normal safe limits. The patient is Critical when the vital signs of the patient are fluctuating and they do not satisfy the normal patient limits. The patient may have, at some point, lost their consciousness. This type of patient usually will require critical care or some other treatment in the hospital’s intensive care unit (ICU). Nevertheless, such a factor as general condition has so far not been taken into account in the analyses of the KD diagnosis. We are aware that the general condition defined as fair is not unique and pathognomonic for KD, but its important feature is that no additional equipment is needed. Moreover, its assessment may help in deciding whether to treat or not because additional symptoms may appear later in the disease or may not occur at all. We believe that further studies should be carried out to optimize and validate this factor in similar analyses.

In addition, these rules are important at the beginning of hospitalization because they consider only one additional factor (clinical symptom), not all the criteria (four out of five required to state a diagnosis of classic KD). This means a possibility of faster diagnosis, especially in the atypical KD. It may reduce the number of complications—coronary aneurysms among them [15]. Kim and Kim presented a study considering the differential diagnosis of Kawasaki Disease based on acute cervical lymphadenopathy using the decision tree method [16]. It took into account radiological (Ultrasonography/Computer tomography) and laboratory findings (CRP, general blood count). The usefulness of such a diagnostic algorithm was shown, but it requires additional tests, especially radiological ones. That is why it may be less useful in every day clinical practice.

It should be added that we included only children with a clinical suspicion of KD and with a definite alternative final diagnosis. The exclusion of other children was necessary to avoid including to non-KD group children who had KD but were not correctly diagnosed. From the hospital perspective, these three clinical entities are the most difficult to distinguish.

Viral infections other than EBV (Epstein-Barr Virus) ones do not have prolonged fever, and children are not very sick, so parents do not bring them to hospital. In invasive bacterial infection, the history of fever is usually short, and because of the severity of the condition, parents go to the hospital immediately. Other diseases such as JRA (juvenile rheumatoid arthritis) and SLE (systemic lupus erythematosus) are very rare, and it was not possible to form a group enough large for comparison.

Half of the decision rules contained inflammatory markers (CRP and ESR). CRP is produced by hepatocytes in response to inflammatory processes. However, its production is not only activated by an infectious agent but also a non-infectious one [17]. Therefore, this is a non-specific marker. CRP was initially used to differentiate infectious diseases (viral versus bacterial etiology of disease), but over time it has been noticed that this protein is useful in estimating the prognosis of inflammatory process (not necessarily infectious diseases), including KD [18,19,20,21,22].

Among the analyzed patients, the CRP concentration was different in each disease. The highest concentration of this marker was observed in patients with KD (mean 112.3 mg/L) compared to infectious diseases (infectious mononucleosis—26.9 mg/L, streptococcal pharyngitis—50.9 mg/L). Statistical significance was demonstrated in the above differences, but in practice, it is a very non-specific indicator and cannot be individually taken into account in the differentiation of the above diseases. However, an advantage of using CRP can be useful by adding the DRSA method. As shown above, this marker combined with the factors obtained from the physical examination resulted in an excellent sensitivity result.

ESR is simple and cheap to perform. However, its variability in both physiological and pathological states and low specificity, together with the introduction of newer and newer inflammatory markers into laboratory practice, make it nowadays less used [23,24,25]. Its increase is observed, as in the case of CRP, in all inflammatory processes; therefore, it seems that if both tests can be performed, CRP should be checked [26]. However, SR is still included in the American Heart Association guidelines for the diagnosis of Kawasaki disease as an additional test [27]. A meta-analysis by Xuan et al. showed that SR may be a prognostic factor for KD resistance to first-line immunoglobulin therapy [28,29]. In the analyzed group of patients, no comparison was made between them due to missing data. In the hospital, the principle of the superiority of CRP over ESR was followed. The group of children with KD (where ESR was measured) also had anemia and hypoalbuminemia, which are the factors that can increase the ESR result themselves. Thus, it can be concluded that SR may be probably useful in analyzing the prognosis of resistance to immunoglobulins used in therapy, but it is not an appropriate differentiating factor. However, using the DRSA method, which enables comparative analyses even in the absence of some data, the combination of thrombocytosis and an increased ESR ≥ 77 mm/h was found to be highly sensitive in the diagnosis of KD.

In addition, the DRSA analysis showed that general abnormalities in the additional tests, such as thrombocytosis or anemia, are not good factors in the differential diagnosis of KD. The typical causes of thrombocytosis are inflammatory diseases (both infectious and non-infectious), postoperative conditions, trauma, burns, blood loss, asplenia or hyposplenia [30]. What is more, anemia is a common problem in the population, also in Poland [31]. Knowing that thrombocytosis is associated with anemia by pathophysiological mechanisms, these factors are not useful in approximating the diagnosis of KD.

To conclude, it is important that the DRSA analysis made it possible to create decision rules with a small amount of data, in some cases incomplete (only 150 patients were analyzed). With the enormous potential of this method, further research should be carried out, especially prospective ones. Observing the increasing frequency of Pediatric Inflammatory Multisystem Syndrome clinically corresponding to KD, the results would be interesting.

## 5. Conclusions

A DRSA analysis may be helpful to diagnose of KD at an early stage of the disease. It can be used even with a small amount or incomplete clinical or laboratory data. The most valuable rules for a KD diagnosis, which were created on the basis of the data presented in this work, are as follows:

If a child with suspected KD has conjunctivitis and CRP ≥ 40.1 mg/L, it is KD.

If a child with suspected KD has thrombocytosis and ESR ≥ 77 mm/h, it is KD.

If a child with suspected KD has a fair general condition and fever ≥ 5 days and rash, it is KD.

If a child with suspected KD has a fair general condition and fever ≥ 5 days and conjunctivitis, it is KD.

If a child with suspected KD has a fever ≥ 5 days and rash and CRP ≥ 7.05 mg/L, this is KD. What is more, we think that it is worth using the DRSA method in medicine. It is useful for differential diagnosis, especially when the disease does not have specific tests to be diagnosed. Much more prospective studies should be performed to confirm this thesis.

## Figures and Tables

**Figure 1 children-08-00929-f001:**
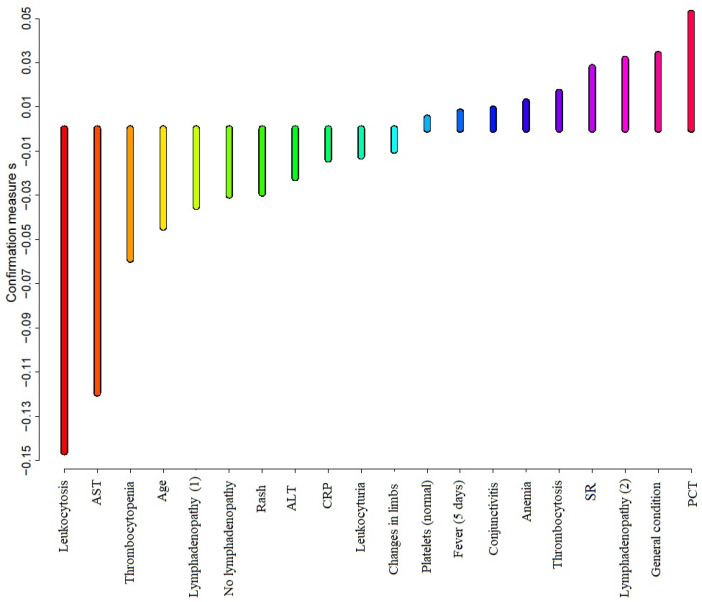
Predictive attribute confirmation calculated for original attributes.

**Table 1 children-08-00929-t001:** Comparison of the physical examination and laboratory findings in Kawasaki disease, infectious mononucleosis and *S. pyogenes* pharyngitis.

Disease/Analysed Feature	Kawasaki Disease	Infectious Mononucleosis	*S. pyogenes* Pharyngitis
Physical examination findings			
Etiology	Unknown	Epstein-Barr Virus (90%)	*S. pyogenes*
Changes in limbs	erythema and/or swelling of the hands and feet flaky peeling of the epidermis	N/A	flaky peeling of the epidermis of hands and feet
Skin changes	a polymorphic non-vesicular rash located mainly on the trunk possible measles-like or scarlet-like character	a petite or lumpy rash, in case of amoxicillin prescription, measles-like character	a petite rash confluent on the trunk, elbows and knees Filatow’s triangle Pastii lines
Eyes	nonexudative bilateral conjunctivitis	swelling of the eyelids	N/A
Changes on the mucous membranes	congestion of the mucous membranes of the mouth and throat, raspberry tongue	congestion of the mucous membranes of the mouth and throat	congestion of the mucous membranes of the mouth and throat, raspberry tongue
Lymph nodes	unilateral enlargement of the cervical lymph nodes	symmetrical enlargement of the cervical lymph nodes	symmetrical enlargement of the cervical lymph nodes
Additional laboratory findings			
Blood count	Erythrocytes ↓ Leukocytes ↑ Platelets ↑	Erythrocytes N Leukocytes ↑/N Platelets ↓/N	Erythrocytes N Leukocytes ↑ Platelets N
ALT	↑↑	↑↑	N
AST	↑	↑	N
CRP	↑↑	↑	↑↑
SR	↑↑	↑	↑↑
PCT	↑↑	N	↑↑
Specific additional tests	N/A	anti-VCA anti-EA anti-EBNA	StrepTest Pharyngeal swab

Abbreviations: ALT—glutamic pyruvic transferase; anti-VCA—anti-viral capsid antigen; anti-EA—anti-early antigen; anti-EBNA—anti-EBV nuclear antigen; AST—aspartate transaminase; CRP—C-reactive protein; SR—Sedimentation Rate; PCT—procalcitonin; StrepTest—Group A Streptococcus antigen immunoassay (throat swab); N—normal; ↓—below normal range; ↑—slightly above normal range; ↑↑—much above normal range.

**Table 2 children-08-00929-t002:** Predictive attribute confirmation calculated for original attributes.

Attribute	f	A	Z	l	c1	s
Leukocytosis	−0.6004	−0.0052	−0.1551	−1.3876	−0.0776	−0.1460
AST	−0.5424	−0.0096	−0.1265	−1.2152	−0.0633	−0.1194
Thrombocytopenia	−0.3471	−0.0252	−0.0610	−0.7243	−0.0305	−0.0588
Age	−0.2678	−0.0679	−0.0428	−0.5490	−0.0214	−0.0443
Lymphadenopathy (1)	−0.2390	−0.0044	−0.0370	−0.4875	−0.0185	−0.0350
Lymphadenopathy (0)	−0.2090	−0.0075	−0.0313	−0.4242	−0.0156	−0.0298
Rash	−0.1946	−0.0324	−0.0287	−0.3941	−0.0143	−0.0289
ALT	−0.1588	−0.0123	−0.0225	−0.3203	−0.0113	−0.0219
CRP	−0.0873	−0.0434	−0.0116	−0.1750	−0.0058	−0.0135
Leukocyturia	−0.0923	−0.0104	−0.0123	−0.1851	−0.0061	−0.0122
Extermities changes	−0.0734	−0.0080	−0.0096	−0.1470	−0.0048	−0.0095
Platelets (norm)	0.0388	0.0006	0.0705	0.0777	0.0353	0.0049
Fever (5 days)	0.0413	0.0032	0.0749	0.0827	0.0374	0.0076
Conjunctivitis	0.0631	0.0022	0.1122	0.1263	0.0561	0.0089
Anemia	0.1040	0.0013	0.1789	0.2087	0.0895	0.0122
Thrombocytosis	0.1413	0.0022	0.2361	0.2845	0.1180	0.0165
SR	0.2998	0.0005	0.4457	0.6186	0.2228	0.0277
Lymphadenopathy (2)	0.3364	0.0018	0.4877	0.7001	0.2439	0.0315
General condition	0.2559	0.0103	0.3924	0.5234	0.1962	0.0336
PCT	0.5073	0.0128	0.6591	1.1182	0.3296	0.0523

Lymphadenopathy (0)—cervical lymph nodes not enlarged, Lymphadenopathy (1)—one side enlargment of cervical lymph nodes; Lymphadenopathy (2)—symmetrical (both sides) enlargment of cervical lymph nodes.

**Table 3 children-08-00929-t003:** General information about the analyzed group.

	Kawasaki Disease	Infectious Mononucleosis	*S. pyogenes* Pharyngitis
Patients (n)	48	49	53
Mean age (min-max in months)	34 (1–136)	36 (12–57)	43 (18–60)
Sex (n)			
Girls	14 (29%)	23 (47%)	24 (45%)
Boys	34 (71%)	26 (53%)	29 (55%)
Hospitalization length (min-max in days)	11 (3–21)	4 (1–10)	4 (1–9)
Fever ≥ 5 days(% n)	48 (100%)	13 (27%)	3 (6%)

## Data Availability

The data presented in this study are available on request from the corresponding author.

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
