# Peer review of "Can AI Help Pediatricians? Diagnosing Kawasaki Disease Using DRSA"

_children, 2021, doi:10.3390/children8100929_

Round 1

Reviewer 1 Report

This paper proposes a dominance-based rough set approach (DRSA) method for finding decision rules for diagnosing Kawasaki disease (KD), infectious mononucleosis and S. pyogenes pharyngitis in children.

I have a number of questions/comments:

1) Line 65: "Age restriction ... we wanted to eliminate adolescent patients with infectious mononu-65 cleosis. Epidemiological data show that this disease is most common in older children"

Please provide references. I'm also not following this justification for excluding older infectious mononucleosis patients. You are comparing your entire KD patient population to a subset of mononucleosis and S. pyogenes patients and establishing a set of decision rules based on this subset. The intended use for these decision rules is to aid in early diagnosis of KD regardless of age ("early stage of the disease"). How can you be confident that your decision rules will perform well on adolescent patients if you did not use adolescent data for two of the three diseases when making the rules?

2) Line 81: "When are these laboratory results collected? Is it the first set of tests after admission?"

When are these laboratory tests collected? Is it the first set of tests after admission?

3) Table 3:

How many Kawasaki Disease (KD) patients were <5 years of age? Since KD mostly occurs in children <5 years old, would the DRSA method produce a similar set of rules if the age restriction that was used for mononucelosis/S. pyrogenes infection was also applied to KD?

4) Please provide more information on how confirmation measure was calculated. How are we supposed to interpret Figure 1? No explanation is given as how to rank feature importance from the confirmation measure values.

5) Line 259: "With the enormous potential of this method, further research should be carried 259 out, especially prospective ones."

It's not clear how or if DRSA is an advantage over other methods (e.g. odds ratios from an ordinal regression method). Please justify the "enormous potential" of DRSA. More generally, the paper lacks proper comparisons to other techniques.

Author Response

Dear Sir/Madam,

thank you for your questions. Let me answer them one by one:

1) Line 65: "Age restriction ... we wanted to eliminate adolescent patients with infectious mononu-65 cleosis. Epidemiological data show that this disease is most common in older children" Please provide references.

One of the articles summarizing infectious mononucleosis: Smatti MK, Al-Sadeq DW, Ali NH  i wsp. Epstein-Barr Virus Epidemiology, Serology, and Genetic Variability of LMP-1 Oncogene Among Healthy Population: An Update. Front Oncol. 2018;8:211. Corrected in the text

I'm also not following this justification for excluding older infectious mononucleosis patients. You are comparing your entire KD patient population to a subset of mononucleosis and S. pyogenes patients and establishing a set of decision rules based on this subset. The intended use for these decision rules is to aid in early diagnosis of KD regardless of age ("early stage of the disease"). How can you be confident that your decision rules will perform well on adolescent patients if you did not use adolescent data for two of the three diseases when making the rules? 

We did not want it to apply to adolescents, because KD epidemiology clearly indicates that it applies to children under the age of 5. That was the basic assumption - hence the age restriction. In addition, the epidemiology of infectious mononucleosis (as above) indicates its occurrence in older children, that is why we did not compare such age-inconsistent age groups.

2) Line 81: "When are these laboratory results collected? Is it the first set of tests after admission?" When are these laboratory tests collected? Is it the first set of tests after admission? 

Blood for laboratory tests was collected during children’s admission. It was to serve for early diagnosis. Corrected in the text

3) Table 3:

How many Kawasaki Disease (KD) patients were <5 years of age? Since KD mostly occurs in children <5 years old, would the DRSA method produce a similar set of rules if the age restriction that was used for mononucelosis/S. pyrogenes infection was also applied to KD?

88% of children with KD were under the age of 5, so we did not perform age restriction on purpose (as above). 

4) Please provide more information on how confirmation measure was calculated. How are we supposed to interpret Figure 1? No explanation is given as how to rank feature importance from the confirmation measure values. 

Estimation of both rule and attribute relevancewas performed by measuring Bayesian confirmation, as described previously. Decision rules were induced repetitively on bootstrap samples and then tested on patients who were not included in the samples. Reported results were obtained in a 5-fold cross validation experiment that was repeated 10 (for a single basic classier) or 100 times. Let us observe that a rule can be seen as a consequence relation “if P, then C,” where P is rule premise, and C rule conclusion. The relevance of a rule is assessed by the Bayesian confirmation measure which quantifies the contribution of rule premise P to correct classification decision of unseen patients. For some reasons described previously, we chose confirmation measure denoted by s(C,P), for its easy interpretation as difference of conditional probabilities involving C and P in the following way:s(C,P)=Pr(CjP)Pr(Cj:P), where probability Pr() is estimated on the testing samples of patients. The relevance of each single attribute is also assessed by the Bayesian confirmation measure,but in this case, it quantifies the degree to which the presence of attribute atri in premise P, denoted by atri P, provides evidence for or against rule conclusion C. Here, we use again confirmation measure s(C,atri P), but now it is defined as follows: s(C,atiP)=Pr(Cjatri P)Pr(Cjatri :P). In consequence, the attributes being present in the premise of rules that make correct decisions,or attributes absent in the premise of rules that make incorrect decisions, become more relevant.

5) Line 259: "With the enormous potential of this method, further research should be carried 259 out, especially prospective ones." It's not clear how or if DRSA is an advantage over other methods (e.g. odds ratios from an ordinal regression method). Please justify the "enormous potential" of DRSA. More generally, the paper lacks proper comparisons to other techniques. 

DRSA enables testing even in the case of incomplete data, enables comparison of parametric and non-parametric data. The disadvantage of classical scaling system is that the conversion of clinical data into numeric values risks losing the primary character of the data. Different parameters are added together as if they were equal, such that the sum of completely different parameters can yield the same results. The obtained results are non-informative, in that they do not show how the diagnosis was made. It discourages the application of this system in therapeutic decisions because it does not give decision-makers the chance to evaluate the independent results. 

Yours sincerely,
Bartosz Siewert

Reviewer 2 Report

Bartosz Siewert et al describe the usefulness of dominance-based rough set approach (DRSA) to diagnosing Kawasaki disease (KD). Because it is still difficult to make an accurate diagnosis of KD, the establishment of a new diagnostic method is needed. The use of AI for this purpose is very unique, and the analysis method and results of this paper are interesting. However, there are some problems or unclear points in this manuscript.

Major comments

1) Because many of the readers of this journal will be clinicians, they may not be familiar with DRSA. More detailed explanations of what this method can reveal or how it can be applied clinically should be added to the Introduction section.

2) In Materials and Methods section, the authors demonstrated that this study was conducted both retrospectively and prospectively. However, they did not show the methods of the retrospective study and prospective study, respectively. Please clearly describe them so that the reader can easily understand.

3) It is unclear the diagnostic criteria of KD. The search of this study was done based on the ICD-10 codes, but the definition of KD is not explicitly mentioned. Were patients diagnosed using the AHA criteria? Also, are these patients including incomplete KD? I think that the diagnostic certainty is a prerequisite for this analysis.

4) In DRSA results, the authors describe that “If a child suspected of KD…” , however there are other clinical conditions of suspected KD other than infection mononucleosis and S. pyogenes pharyngitis. I don’t think this analysis cover all suspected cases of Kawasaki disease.

5) Since the laboratory data changes over time in KD, please indicate which timing of the value the authors used. In addition, there is no description of the cut-off value for each parameter. It is also unclear how “general condition” is evaluated.

6) There is no information about informed consent.

Minor comments

1) Abbreviations (AI, DRSA) should be avoided in title.

2) What do the numbers on the far left of Table 2 represent?

Author Response

Dear Sir/Madam,

thank you for your questions. Let me answer them one by one:

1) Because many of the readers of this journal will be clinicians, they may not be familiar with DRSA. More detailed explanations of what this method can reveal or how it can be applied clinically should be added to the Introduction section. 

DRSA enables testing even in the case of incomplete data, enables comparison of parametric and non-parametric data. The disadvantage of classical scaling system is that the conversion of clinical data into numeric values risks losing the primary character of the data. Different parameters are added together as if they were equal, such that the sum of completely different parameters can yield the same results. The obtained results are non-informative, in that they do not show how the diagnosis was made. It discourages the application of this system in therapeutic decisions because it does not give decision-makers the chance to evaluate the independent results. Added in the text. 

2) In Materials and Methods section, the authors demonstrated that this study was conducted both retrospectively and prospectively. However, they did not show the methods of the retrospective study and prospective study, respectively. Please clearly describe them so that the reader can easily understand.

We mean the way of data collection. The analysis started in early 2019, when patient data was collected retrospectively (2015-2018). Then, throughout 2019, each patient with definitive diagnosis of KD was included in the analyzed group. It is true that it was actually also retrospective as the patient was included after diagnosis. Corrected in the text.

3) It is unclear the diagnostic criteria of KD. The search of this study was done based on the ICD-10 codes, but the definition of KD is not explicitly mentioned. Were patients diagnosed using the AHA criteria? Also, are these patients including incomplete KD? I think that the diagnostic certainty is a prerequisite for this analysis.
4) In DRSA results, the authors describe that “If a child suspected of KD…” , however there are other clinical conditions of suspected KD other than infection mononucleosis and S. pyogenes pharyngitis. I don’t think this analysis cover all suspected cases of Kawasaki disease. 

Answer for 3 and 4: Kawasaki disease can be suspected in any patient with prolonged fever, pharyngitis. The inclusion criteria was clinical suspicion of Kawasaki disease. We analysed only patients with definite diagnosis. That is why we used ICD-10 codes. The definite diagnosis was based on AHA criteria. Patients, whith not established diagnosis were excluded from the analysis. Corrected in the text.

5) Since the laboratory data changes over time in KD, please indicate which timing of the value the authors used. In addition, there is no description of the cut-off value for each parameter. It is also unclear how “general condition” is evaluated. 

Blood for laboratory tests was collected during children’s admission. It was to serve for early diagnosis. Corrected in the text
The values ​​of the blood count parameters and the activity of ALT and AST enzymes as well as the deviations in the results of these tests were analyzed depending on the norm defined for a given age group. Venous blood counts were based on data published in Pediatric reference intervals.
Soldin SJ, Wong EC, Brugnara C i wsp. Pediatric reference intervals. Washington, AACC Press, 2011
Lockitch G, Halstead AC, Albersheim S i wsp. Age- and sex-specific pediatric reference intervals for biochemistry analytes as measured with the Ektachem-700 analyzer. Clin. Chem., 1988; 34 (8): 1622–1625
Schwimmer JB, Dunn W, Norman GJ i wsp. SAFETY study: alanine aminotransferase cutoff values are set too high for reliable detection of pediatric chronic liver disease. Gastroenterology, 2010; 138: 1357–1364

Assessment of the general condition was performed by the trained medical personnel - doctors. Additionally, they were authorized by the Head of the Department to work individually on shifts.

6) There is no information about informed consent.

In Poland, for this type of work, it is not necessary to obtain a decision of the Bioethics Committee or informed consent, because it is not a medical experiment. This work was based on clinical data obtained from the hospital's IT system, for which consent was obtained from the Hospital Management.

Minor comments

1) Abbreviations (AI, DRSA) should be avoided in title.
I agree, we should definitely not do this, but due to the limitations of the number of characters in the title and the desire to convey as much information as possible, we decided to create such a title.

2) What do the numbers on the far left of Table 2 represent?
These are the numbers (order) of the analyzed parameters.

Yours sincerely,
Bartosz Siewert

Round 2

Reviewer 2 Report

The authors have answered the questions one by one, however, some of the questions have not been answered adequately.

Comments

1) The authors mentioned that the inclusion criteria of this study were clinical suspicion of Kawasaki disease. However, not only infection mononucleosis and S. pyogenes pharyngitis, there are also other diseases that should be differentiated from KD. Because the authors used ICD-10 and excluded suspected Kawasaki disease cases that had not been diagnosed, “If a child suspected of KD…” seems to be overstated. They need to either correct this statement “If a child suspected of KD” or add this limitation in Discussion section.

2) Regarding the laboratory data, specify the timing of taking blood samples, such as at diagnosis, before treatment, or after treatment, rather than only “during children’s admission”

3) Regarding “general condition”, it is still unclear how the authors categorize it, that is, only good or bad. In other words, it is not clear which are continuous variables or categorical variables including other parameters. The authors should indicate how the parameters were categorized if they were categorical variables, for example, the definition of leukocytosis, thrombocytopenia, anemia, or general condition.

4) In general, the use of clinical information requires informed consent even though it is a retrospective study (opt-out form can be acceptable). Therefore, the information about informed consent must be mentioned on the manuscript. The authors should describe about the rule of informed consent for this type of study and the permission of hospital management.

5) Regarding the numbers on the far left of Table 2, please clarify on the table if the order of the analyzed parameters is important. If not, these numbers must be removed.

6) Abbreviations must be listed in each table, for example, Erythrocytes N, Leukocytes/N, Platelets N, ALT, AST, CRP, OB, PCT,VCA, EA, EBNA, SR.

Author Response

1) The authors mentioned that the inclusion criteria of this study were clinical suspicion of Kawasaki disease. However, not only infection mononucleosis and S. pyogenes pharyngitis, there are also other diseases that should be differentiated from KD. Because the authors used ICD-10 and excluded suspected Kawasaki disease cases that had not been diagnosed, “If a child suspected of KD…” seems to be overstated. They need to either correct this statement “If a child suspected of KD” or add this limitation in Discussion section.

We included only children with a clinical suspicion of KD and with a definite alternative final diagnosis. Exclusion of other children was necessary to avoid including to non-KD group children who had KD but was not correctly diagnosed. From the hospital perspective these three clinical entities are the most difficult to distinguish.
Viral infections other than EBV do not have prolonged fever and children are not very sick so parents do not bring them to hospital. In invasive bacterial infection the history of fever is usually short and because of the severity of the condition parents go to the hospital immediately. Other diseases such as JRA, SLE are very rare and it was not possible to form a group enough large for comparison.
Corrected in the text - discussion

2) Regarding the laboratory data, specify the timing of taking blood samples, such as at diagnosis, before treatment, or after treatment, rather than only “during children’s admission”

It was during children’s admission so at the beggining of diagnostic process, also before treatment started.
Corrected in the text - materials and methods

3) Regarding “general condition”, it is still unclear how the authors categorize it, that is, only good or bad. In other words, it is not clear which are continuous variables or categorical variables including other parameters. The authors should indicate how the parameters were categorized if they were categorical variables, for example, the definition of leukocytosis, thrombocytopenia, anemia, or general condition.

Assessment of general condition was performed by the trained medical personnel - doctors. This is a subjective assesment performed as a part of routine examination of each patient – and as such recorded in patient medical records. Terms used to describe general condition of patient commonly used are: good, fair, serious, critical. The American Hospital Association has advised physicians to use general terminology when describing their patient’s state of health. Good That patient’s vital signs are stable and they are within the prescribed normal limits. The patient is conscious and the patient feels comfortable. Fair The patient’s vital signs may be stable and within the prescribed normal limits. While the patient may be conscious, there may however, be minor complications, making the patient somewhat uncomfortable. Serious The patient’s vital signs might be fluctuating and they do not adhere to normal safe limits. Critical The vital signs of the patient are fluctuating and they don’t satisfy the prescribed normal patient limits. The patient may have at some point lost their consciousness. This type of patient usually will require critical care, or some other treatment in the hospital’s intensive care unit (ICU).
Corrected in the text - discussion

4) In general, the use of clinical information requires informed consent even though it is a retrospective study (opt-out form can be acceptable). Therefore, the information about informed consent must be mentioned on the manuscript. The authors should describe about the rule of informed consent for this type of study and the permission of hospital management.

On admission to hospital all patients’ guardians are informed that medical data obtained during hospital staying may be anonymously used for various analysis. That kind of consent is enough for such kind of research from ethical point of view in Poland.
Corrected in the text - informed consent statement

5) Regarding the numbers on the far left of Table 2, please clarify on the table if the order of the analyzed parameters is important. If not, these numbers must be removed.

Not important – corrected (removed) in the text. (Table 2)

6) Abbreviations must be listed in each table, for example, Erythrocytes N, Leukocytes/N, Platelets N, ALT, AST, CRP, OB, PCT,VCA, EA, EBNA, SR.

Corrected in the text (Table 1)

This manuscript is a resubmission of an earlier submission. The following is a list of the peer review reports and author responses from that submission.